# Steady-State Levels of Miro1 Linked to Phosphorylation at Serine 156 and Mitochondrial Respiration in Dopaminergic Neurons

**DOI:** 10.3390/cells11081269

**Published:** 2022-04-08

**Authors:** Lisa Schwarz, Julia C. Fitzgerald

**Affiliations:** Department of Neurodegeneration, Hertie Institute for Clinical Brain Research, University of Tuebingen, 72076 Tuebingen, Germany; l.schwarz@uni-tuebingen.de

**Keywords:** Miro1, PINK1, Parkinson’s disease, mitochondria

## Abstract

Miro1 has emerged as an interesting target to study Parkinson’s disease-relevant pathways since it is a target of PINK1 and Parkin. Miro1 is a mitochondrial GTPase with the primary function of facilitating mitochondrial movement, and its knockout in mice is postnatally lethal. Here, we investigated the effect of the artificial *RHOT1*/Miro1 S156A mutation since it is a putative PINK1 phosphorylation site shown to be involved in Miro1 degradation and mitochondrial arrest during mitophagy. We gene-edited a homozygous phospho-null Miro1 S156A mutation in induced pluripotent stem cells to study the mutation in human dopaminergic neurons. This mutation causes a significant depletion of Miro1 steady-state protein levels and impairs further Miro1 degradation upon CCCP-induced mitophagy. However, mitochondrial mass measured by Tom20 protein levels, as well as mitochondrial area, are not affected in Miro1 S156A neurons. The mitochondria are slightly lengthened, which is in line with their increased turnover. Under basal conditions, we found no discernable effect of the mutation on mitochondrial movement in neurites. Interestingly, the S156A mutation leads to a significant reduction of mitochondrial oxygen consumption, which is accompanied by a depletion of OXPHOS complexes III and V. These effects are not mirrored by Miro1 knockdown in neuroblastoma cells, but they are observed upon differentiation. Undifferentiated Miro1 S156A neural precursor cells do not have decreased Miro1 levels nor OXPHOS complexes, suggesting that the effect of the mutation is tied to development. In mature dopaminergic neurons, the inhibition of Miro1 Ser156 phosphorylation elicits a mild loss of mitochondrial quality involving reduced mitochondrial membrane potential, which is sufficient to induce compensatory events involving OXPHOS. We suggest that the mechanism governing Miro1 steady-state levels depends on differentiation state and metabolic demand, thus underscoring the importance of this pathway in the pathobiology of Parkinson’s disease.

## 1. Introduction

The atypical GTPase Miro1 is located at the outer mitochondrial membrane and possesses two central EF-hand calcium-binding domains with two surrounding GTPase domains [1,2,3]. Its primary function is facilitation of mitochondrial movement by anchoring mitochondria via the adaptor proteins TRAK1/2 to molecular motors such as kinesin [2,4,5].

Mammalian cells have two Miro proteins exhibiting similar functions, but they are non-redundant [1,6]. Miro1 knockout is postnatally lethal in mice, while Miro2 knockout is tolerated [7]. Hippocampal cultures from Miro2 knockout mice exhibit unaltered mitochondrial trafficking, while this is markedly reduced in Miro1 knockout neurons [6]. These findings highlight the importance of Miro1 in neurons, as well as the necessity of proper mitochondrial distribution for neuronal development and health. Since mitochondria are mostly synthesized in the soma, they need to be distributed to the cell’s periphery [8]. Mitochondrial stopping is also mediated by Miro1 through calcium binding upon elevation of cytosolic calcium [9,10,11,12,13], which is thought to be important for the arrest of mitochondria at sites of high energy demand [8].

Miro1 takes part in several other mitochondrial functions, including mitochondrial calcium buffering and mitochondrial quality control. Miro1 interacts with PINK1 [14] and Parkin [15]; mutations in the encoding genes *PINK1* and *PARK2* cause autosomal recessive Parkinson’s disease [16,17,18]. PINK1 initiates mitophagy upon mitochondrial depolarization and recruits Parkin for the ubiquitination of a subset of mitochondrial substrates [19,20]. The subsequent phosphorylation of the ubiquitin chains by PINK1 activates autophagy [21,22]. The phosphorylation of Miro1 by PINK1, followed by its ubiquitination by Parkin, induces Miro1 removal from depolarized mitochondria and degradation by the proteasome [15,23,24,25]. PINK1 and Parkin also inhibit mitochondrial movement upon mitochondrial depolarization by untethering mitochondria from the microtubules [23,26].

In the context of PINK1-mediated degradation, three residues have been identified for Miro1 phosphorylation (Ser156, Thr298, and Thr299), of which Ser156 was found to be required for degradation [23,26]. However, studies investigating the significance of Ser156 phosphorylation in induced mitophagy are contradictory. While some studies indicate that phosphorylation at Ser156 facilitates Miro1 degradation [23,26], other studies were not able to reproduce this result [15,24]. Because these studies focused on Miro1 degradation, it is not known whether this phosphorylation site has other functions. One limitation was that these studies used the expression of constructs that only show transient effects with the wild-type protein still present. The mechanisms governing Miro1 retention at or degradation at mitochondria are of particular relevance: a study shows that a large subset of Parkinson’s disease patients’ fibroblasts fail to remove Miro1 following mitochondrial depolarization [27].

We previously established induced pluripotent stem cell (iPSC) lines where we introduced *RHOT1*/Miro1 mutations, including S156A, into healthy iPSCs using CRISPR/Cas9 gene editing [28].

Here, we show the effect of the phospho-null mutant S156A in human dopaminergic neurons, the cell type affected in Parkinson’s disease. Our results confirm the findings of previous studies, in which preventing Miro1 phosphorylation at this site affects Miro1 degradation, slowing the flux of mitophagy in dopaminergic neurons. This is accompanied by mild, but consistently reduced, mitochondrial membrane potential, basal respiration, and OXPHOS complex levels without any effect on respiratory capacity or mitochondrial movement. This only occurs in differentiated neurons which cannot rely on canonical mitophagy.

## 2. Materials and Methods

### 2.1. Generation Neural Precursor Cells (NPCs) and Differentiation in Midbrain Dopaminergic Neurons (hDaNs)

NPCs were derived from iPSCs as previously described [29] with the following adjustments: EBs were started in base medium (50% DMEM/F12 (Thermo Fisher Scientific (Waltham, MA, USA), #11-330-057), 50% neurobasal (Thermo Fisher Scientific (Waltham, MA, USA), 21103-049), 1% penicillin/streptomycin (Merck (Darmstadt, Germany), #A2213), 1% GlutaMax (Thermo Fisher Scientific (Waltham, MA, USA), #35050-038), 1% B27 supplement (without vitamin A; Thermo Fisher Scientific (Waltham, MA, USA), #12587-010), and 0.5% N2 supplement (Thermo Fisher Scientific (Waltham, MA, USA), #17502-048)), supplemented with 20 µM SB431542 (Selleckchem (Houston, TX, USA), #S1067-10 mg), 1 µM dorsomorphin (Axon Medchem (Groningen, The Netherlands), #Axon 1708 5 mg), 3 µM CHIR 99021 (Axon Medchem (Groningen, Netherlands), #Axon1386 5 mg), and 0.5 µM PMA (Merck (Darmstadt, Germany), #540220-5MG). On day 3, EBs were put in NPC medium (base medium supplemented with 3 µM CHIR 99021, 200 µM ascorbic acid (Sigma-Aldrich (St. Louis, MO, USA), #A4544-25G), and 0.5 µM PMA) and plated on Matrigel (Corning (Corning, NY, USA), #354230) on day 5. After several passages, NPCs were differentiated into human, mid-brain-specific dopaminergic neurons as previously described [30], with the alteration that NPCs were not primed by removing the PMA from the NPC media prior to differentiation. Dopaminergic neurons were used at day 16–19. The number of individual differentiations used in experiments is indicated as n_Diff_. 

For the induction of mitophagy, hDaNs were treated for 2, 4, 6, or 24 h with a final concentration of 10 µM CCCP in the neuronal maturation medium, or for 6 h with 10 µM CCCP plus 10 µM MG132. Untreated hDaNs received fresh neuronal maturation medium for 24 h.

### 2.2. RT-qPCR

RNA was isolated using the RNEasy kit (Qiagen (Hilden, Germany), #74104), following the manufacturer’s instructions, including the homogenization step using QIAshredder. For RT-qPCR, the Quantitect SYBR-green kit (Qiagen (Hilden, Germany), #204243) was used, following the manufacturer’s instructions. Primers used for RT-qPCR are listed in Table 1.

### 2.3. Immunofluorescence Staining

NPCs and hDaNs were seeded on Matrigel-coated coverslips and fixed with 4% PFA in PBS. For nuclear marker Sox2, cells were permeabilized with neat methanol for 5 min at −20 °C. For mitochondrial marker Tom20, cells were permeabilized with 0.5% TritonX-100 in PBS for 5 min at room temperature. After washing with wash buffer (0.01% TritonX-100 in PBS), cells were blocked using 10% normal goat serum in wash buffer. Antibodies were diluted in 5% normal goat serum in wash buffer. Primary antibodies (MAP2 1:2000 (abcam (Cambridge, UK), #ab5392), Tyrosine Hydroxylase (TH) 1:1000 (Pel-Freez (Rogers, AR, USA) #P40101-150), Tom20 1:200 (Santa Cruz Biotechnology (Dallas, TX, USA), #sc17764), Nestin 1:150 (R&D systems (Wiesbaden, Germany), #MAB1259), and Sox2 1:1000 (abcam (Cambridge, UK), #ab97959)) were incubated overnight at 4 °C, and secondary antibodies (1:1000; Thermo Fisher Scientific (Waltham, MA, USA), #A21449, #A11070, #A21463, #A11010) for 1–2 h at room temperature. Nuclei were counterstained with Hoechst (Molecular Devices (Urstein, Austria), #H3569). Coverslips were mounted on glass slides using Dako fluorescent mounting medium (Aligent (Santa Clara, CA, USA), #S3023). Slides were imaged using a Zeiss Imager.Z1 equipped with an ApoTome.2 and an AxioCam MRm. Representative images were processed using ZEN software (blue edition, Zeiss (Oberkochen, Germany)). For representative images of MAP2/TH and Nestin/Sox2 staining, Z-stacks were projected (maximum intensity), brightness and contrast were adjusted, and a scale bar was inserted.

### 2.4. SDS-PAGE and Western Blotting

For cell lysis, a lysis buffer containing 1% Triton X-100 in PBS with cOmplete protease inhibitor (Sigma-Aldrich (St. Louis, MO, USA), #11873580001) and PhosStop phosphatase inhibitor (Sigma-Aldrich (St. Louis, MO, USA), #4906837001) were used. hDaN and differentiated SH-SY5Y lysates were homogenized using 25G (5 passes) and 27G (10 passes) needles. SH-SY5Y lysates were homogenized by repetitive vortexing. NPCs were homogenized using 20G (5 passes), 25G (5 passes), and 27G (10 passes) needles. Electrophoresis and immunoblotting onto PVDF membranes (MilliporeSigma (Burlington, MA, USA), #IPVH00010) were done following standard protocols. Primary antibodies were diluted in 5% milk or Roche blocking solution diluted in Tris-buffered saline containing 0.05% Tween20 (TBST). The antibodies used in this study are as follows: α-vinculin 1:5000 (Sigma-Aldrich (St. Louis, MO, USA), #V9131), α-tubulin 1:5000 (Sigma-Aldrich (St. Louis, MO, USA), #AA13), Miro1 1:500 (Thermo Fisher Scientific (Waltham, MA, USA), #PA-42646), Tom20 1:1000 (Santa Cruz Biotechnology (Dallas, TX, USA), #sc11415), Total OXPHOS 1:1000 (Abcam (Cambridge, UK), #ab110413), Mitofusin 1:1000 (Abcam (Cambridge, UK), ab#57602) and LC3 1:1000 (Novus Biologicals (Littleton, CO, USA), #NB100-2220). Fluorescent secondary antibodies were diluted in 5% milk in TBST (1:10,000; LI -COR Biosciences, (Lincoln, NE, USA), #926-32210, #926-32213, #926-68071, and #926-68070). Proteins on the membranes were detected using an Odyssey CLx infrared imaging system (LI -COR Biosciences, (Lincoln, NE, USA)) with Image Studio software (LI -COR Biosciences, (Lincoln, NE, USA)). Intensity of protein bands was quantified using Image Studio Lite Ver 5.2 (LI -COR Biosciences, (Lincoln, NE, USA)). Intensity of the protein bands was normalized to the intensity of the loading control protein band on the same membrane (α-tubulin or α-vinculin).

### 2.5. Imaging Analysis of Mitochondrial Morphology and Membrane Potential

Approximately 150,000 hDaNs were seeded onto Matrigel-coated 35 mm ibidi dishes two to three days prior to imaging. Two dishes were seeded per differentiation, and imaging was conducted on two consecutive days (n_Diff_ = 3; 2 days of imaging per line, 10 images per day). Prior to imaging, hDaNs were stained with 100 nM MitoTracker green (Thermo Fisher Scientific (Waltham, MA, USA), #M7514) and 25 nM Image-iT™ TMRM reagent (Thermo Fisher Scientific (Waltham, MA, USA), #I34361) in maturation medium and incubated for 7 min at 37 °C with 5% CO_2_. For imaging, medium was replaced with maturation medium without phenol red (Neurobasal: Thermo Fisher Scientific (Waltham, MA, USA), #12348017; DMEM/F12: Thermo Fisher Scientific (Waltham, MA, USA), #21041025). Z-stacks were taken with a slice interval of 0.88 µm using a Leica DMi8 Microscope (40× objective) with the LASX software under a controlled environment (37 °C with 5% CO_2_).

Images were analyzed using Fiji (Rasband, W.S., ImageJ, U. S. National Institutes of Health, Bethesda, MD, USA). Z-stacks were projected using average intensity and all slices. Images were converted to 8-bit greyscale.

Mitochondrial morphology was analyzed as previously described [31] with alterations. In detail, greyscale images of MitoTracker green channel were convolved (process–filters–convolve; Kernel: 0 0 −1 0 0/0 −1 −1 −1 0/−1 −1 24 −1 −1/0 −1 −1 −1 0/0 0 −1 0 0), the background was subtracted (process—subtract background: rolling ball radius 10 px), and local contrast was enhanced (process—enhance local contrast (CLAHE): blocksize 9, maximum slope 4, fast unselected). Next, the tubeness tool (plugins–analyze–tubeness: sigma = 0.24156, use calibration information) was used, and images were despeckled (process–noise–despeckle) and converted into 8-bit. The processed images were converted into a binary image by applying thresholding (a similar threshold between images was maintained). An analysis of area, shape descriptors, perimeter, and fit ellipse as measurements was performed on particles bigger than 1.5 px (circularity: 0–1). The output of measurement was further processed in Excel to calculate descriptors: Area [µm^2^]: area; form factor (degree of branching): $\frac{perimeter^{2}}{4\pi \cdot area}$; length [µm]: aspect ratio; circularity: circularity; for each descriptor, the mean of one image was calculated and used as data point in statistics.

For the analysis of mitochondrial membrane potential, MitoTracker green images were processed by subtracting the background (process—subtract background: rolling ball radius 15 px). Next, images were converted to a binary image using thresholding (a similar threshold between images was maintained). A binary image was used as a mask to quantify the integrated density of both the MitoTracker green and TMRM channel (set measurements: integrated density, redirect to MitoTracker green/TMRM greyscale image; analyze particles: size 1.5 px—infinity; circularity: 0–1; outlines were saved). The background was measured for each channel and corrected total fluorescence (CTF) for each particle analyzed was calculated as follows: CTF = IndDen_Particle_ − (area_Particle_ × mean grey value_Background_). The mean CTF of one image was used as a data point for statistics.

### 2.6. Imaging Analysis of Mitochondrial Movement

Approximately 100,000 hDaNs were seeded onto 35 mm ibidi dishes double-coated with poly-L-ornithine (Sigma-Aldrich (St. Louis, MO, USA), #P8638-25MG) and Matrigel three to four days prior to imaging. Two dishes were seeded per differentiation, and imaging was conducted on two consecutive days (n_Diff_ = 3; 2 days of imaging per genotype, 8–9 time-lapses per day). Prior to imaging, hDaNs were stained with 100 nM MitoTracker green in maturation medium and incubated 7 min at 37 °C with 5% CO_2_. For imaging, the medium was replaced with maturation medium without phenol red. Time lapses were taken in 5 s intervals for 5 min (61 frames) using a Leica DMi8 Microscope (40× objective) with the LASX software under a controlled environment (37 °C with 5% CO_2_).

Images were analyzed and processed using Fiji. Time lapses were checked for cellular drift by overlay of the first and last frames. Per imaging day, 10 immobile processes were selected for analysis and subsequently cropped and straightened using plugin straighten (line width: 30 px, applied to entire stack; [32]). The orientation of each straightened process was maintained to have anterograde movement from left to right.

For manual analysis, kymographs were generated by re-slicing (default settings) the straightened processes, then Z-projecting the stack (max intensity) and inverting the LUT; brightness and contrast were adjusted prior to analysis. Tracks were analyzed using the line tool to connect the start and the end of each track. Displacement was calculated using the Pythagorean theorem and mitochondria were classified as stationary (displacement ≤ 2 µm), mobile (displacement > 2 µm), or oscillating (displacement ≤ 2 µm; amplitude (from displacement line to peak of largest amplitude) > 1.5 µm). For each imaging day, the sum of all tracks analyzed was used to calculate fractions of motile, oscillating, anterograde, and retrograde mitochondria; mean displacement of motile mitochondria was calculated from all tracks of one imaging day.

For analysis of mean mitochondrial speed, automated analysis using the TrackMate [33] plugin was performed. First, straightened processes were processed using bleach correction (histogram matching) and enhancement of contrast (process—enhance contrast: 2.5%, normalized, applied to all slices). For TrackMate analysis, the calibration information was used, then the LoG detector was chosen. Next, the detector was set for an estimated blob diameter of 1 µm with threshold 0 and sub-pixel localization. Initial thresholding was not applied, HyperStack Displayer was used, and spots were filtered using the automatic settings (threshold was adjusted manually when automated detection was not appropriate). For tracking of particles, the Linear motion LAP tracker was used with an initial and subsequent search radius of 5 µm, and the max frame gap was set to two frames. The mean average speed of all tracks analyzed per process was used as one data point for statistics.

### 2.7. Imaging Analysis of Mitochondrial Turnover Using MitoTimer Construct

Approximately 80,000 hDaNs seeded on Matrigel-coated glass coverslips were transfected with pMitoTimer plasmid (Addgene (Watertown, MA, USA) plasmid #52659; RRID: Addgene_52659; [34]) using Fugene HD transfection reagent (Promega (Madison, WI, USA), #E2311), according to the manufacturer’s instructions. hDaNs received fresh maturation medium 1–2 h prior to transfection. hDaNs were fixed 48 h after transfection and stained with Tom20 as mitochondrial marker, as described above. Slides were imaged using a Zeiss Imager.Z1 equipped with an ApoTome.2 and an AxioCam MRm. Z-stacks were taken with a 0.72 µm interval with fixed exposure times (Hoechst: 15 ms; green: 150 ms; red: 50 ms; Tom20: 1000 ms). Per differentiation (n_Diff_ = 3), 14 images were analyzed.

Images were analyzed using Fiji. Z-stacks were projected using average intensity and all slices. Images were converted to 8-bit greyscale. Tom20 greyscale images were converted to binary images using thresholding and used as a mask to quantify fluorescence intensity of green and red channels (set measurements: mean grey value, redirect to green or red channel; analyze particles: size 1.5 px—infinity, circularity 0–1, outlines were saved). The mean grey value of all particles from one image was used to calculate the ratio between green and red channels for that image, which was used as one data point for statistics.

### 2.8. Flow Cytometry

Dissociated hDaNs using Accutase (PAN-Biotech (Aidenbach, Germany), #P10-21100) were stained with a final concentration of 100 nM MitoTracker deep red (Thermo Fisher Scientific (Waltham, MA, USA), #M22426) and 100 nM Image-iT™ TMRM for 15 min at 37 °C. hDaNs were measured in neuronal maturation medium without phenol red using a MACSQuant (Mytenyi Biotec (Bergisch Gladbach, Germany)). The mean average fluorescence signal of each channel was divided by the background fluorescence of unstained hDaNs in the same channel to account for autofluorescence.

### 2.9. Respiratory Analyses

For respiratory analyses, a basic mitochondrial stress test measuring oxygen consumption rate (OCR) and extracellular acidification rate (ECAR) was performed using a Seahorse XF96 Extracellular Flux Analyzer (Agilent (Santa Clara, CA, USA)). Approximately 60,000–120,000 neurons were seeded per well into a Matrigel-coated Seahorse cell plate 2–5 days prior to the experiment. For the measurement, the medium was replaced with DMEM base medium (without phenol red; Sigma-Aldrich (St. Louis, MO, USA), #D5030 1L) supplemented with 4.5 mg/mL D-Glucose, 0.22 mg/mL pyruvate, and 2 mM glutamine. After eight measurements of basal respiration, hDaNs were challenged with subsequent injections to give final concentrations of (a) 0.8 µM Oligomycin, (b) 2.7 µM CCCP, and (c) 0.8 µM Rotenone with 4 µM Antimycin A. Each injection was followed by three measurements. OCR and ECAR measurement were normalized to the number of hDaNs seeded for each experiment. Basal respiration was calculated by subtracting the mean of the OCR measurements after injection (c) from the OCR of the third measurement of baseline. Maximal respiration was calculated by subtracting the mean of the OCR measurements after injection (c) from the mean of the OCR measurements after injection (b). Spare respiratory capacity was calculated as the percentage of maximal respiration of basal respiration.

### 2.10. Miro1 Knockdown in Undifferentiated SH-SY5Y

SH-SY5Y cells were maintained in DMEM/F12 supplemented with 10% heat-inactivated FBS, 1% NEAA (Thermo Fisher Scientific (Waltham, MA, USA), #1140-035), and 1% penicillin/streptomycin. For siRNA transfection, cells were seeded 24 h prior to transfecting three consecutive days before harvest and three days after the first transfection (representative blot Miro1 and Tom20). The protocol was modified by transfecting the cells once and harvesting them three days later. Transfection with non-targeting siRNA (Dharmacon (Lafayette, CO, USA) SMART Pool; #D-001206-14-05) and *RHOT1* siRNA (Dharmacon (Lafayette, CO, USA) SMART Pool, #M-010365-01-0005) was performed using 25 ng siRNA with 2 µL transfection reagent (Dharmacon (Lafayette, CO, USA), #T-2001-01) per well, following the manufacturer’s instructions.

### 2.11. Miro1 Knockdown in Differentiated SH-SY5Y

For differentiation, SH-SY5Y cells were seeded on Matrigel. Differentiation was induced by adding DMEM/F12 supplemented with 10 µM all-trans retinoic acid, 1% heat inactivated FBS, 1% NEAA, and 1% penicillin/streptomycin. At day 4 of differentiation, SH-SY5Y were transfected with 50 ng non-targeting or *RHOT1* Dharmacon SMART Pool siRNA (described above) and 4 µL transfection reagent per well with antibiotic-free medium. The medium was replaced with fresh differentiation medium the next day. Differentiated SH-SY5Y were harvested at day 7 of differentiation (three days after transfection).

### 2.12. Statistical Analysis

GraphPad Prism 9 (ver. 9.1.1) was used for statistics. Data are shown as mean ± standard deviation (SD) and were tested for (log)normal using the Shapiro–Wilk test. Normally distributed data were analyzed using a paired *t* test (one-tailed or two-tailed) or 2way ANOVA with Tukey’s multiple comparisons. Non-normally distributed data were analyzed using (multiple) Wilcoxon tests, Mann-Whitney test, or Friedman’s test with Dunn’s multiple comparisons. Experiments with hDaNs were performed using individual differentiations indicated as n_Diff_. The correlation between Miro1 and Complexes V/IV/III (detected on the same membrane) was done in Prism by computing Pearson correlation coefficients and subsequent simple linear regression.

## 3. Results

### 3.1. Miro1 S156A Protein Levels Are Reduced in Dopaminergic Neurons

For investigating Miro1 S156A in dopaminergic neurons, we used previously established, gene edited, homozygous Miro1 S156A iPSCs. We showed that the gene editing did not result in unintended genomic damage and that Miro1 S156A iPSCs were isogenic to the maternal control iPSCs [28].

For differentiation of mid-brain-patterned dopaminergic neurons, we first derived neural precursor cells (NPCs) from Miro1 S156A [28] and isogenic control [35] iPSCs. We validated the identity of the NPCs using RT-qPCR and immunofluorescence (Appendix A) before differentiation into hDaNs. Dopaminergic identity and neuronal maturity of hDaNs were validated showing expression of Tyrosine Hydroxylase (TH) and MAP2, respectively, using RT-qPCR (Figure 1A) and immunofluorescence staining (Figure 1B).

Because previous studies found that phosphorylation at Ser156 is required for Miro1 degradation during induced mitophagy [23,26], we first asked whether the mutation might affect steady-state protein levels. Mean Miro1 protein levels are significantly decreased in Miro1 S156A hDaNs to 58% of the isogenic control (Figure 1C). The reduction of protein levels could be explained by a concomitant reduction of mitochondrial mass, but Tom20 protein levels used as a marker revealed no difference between Miro1 S156A and isogenic control hDaNs (Figure 1D). To determine whether the reduction of Miro1 protein levels is due to a reduction in gene expression, we assessed mRNA levels of *RHOT1*. We also probed *RHOT2* expression to assess compensatory upregulation of Miro2. Both *RHOT1* and *RHOT2* are expressed similarly in Miro1 S156A hDaNs and the isogenic control (Figure 1E). In summary, Miro1 S156A results in significantly reduced Miro1 protein that is not due to a change in mitochondrial mass or gene expression. 

This indicates that S156A either has a destabilizing effect on steady-state levels of Miro1 or elicits a compensatory mechanism in mature neurons to reduce Miro1 protein.

In yeast and drosophila, Gem1/dMiro was shown to regulate mitochondrial morphology via its N-terminal GTPase domain [1,36,37], and a study in rat hippocampal neurons showed that this domain also confers this effect in mammalian cells [4]. Because Ser156 is located in the N-terminal GTPase domain, we assessed whether mitochondrial morphology is affected. We found that in Miro1 S156A hDaNs, mitochondrial morphology is mildly changed by a slight increase in area but significant elongation of mitochondria. Mitochondrial fragmentation and branching were not altered (Figure 1F). Since Miro1 protein is reduced in S156A hDaNs, we next assessed mitochondrial movement. Analysis of kymographs generated from neurites of live neurons (Figure 1G) revealed that there are no differences in the number of mitochondria in motion, nor in the directionality of movement nor the distance that mitochondria travel (Figure 1H). Assessing whether Ser156 is important for regulating the speed of mitochondria under basal conditions, we found that Miro1 S156A mitochondria travel as fast as isogenic control mitochondria (Figure 1I). These findings are in line with previous data that show that S156A alone is not sufficient to increase mitochondrial movement. In stimulated conditions, however, preventing phosphorylation at Ser156 impairs mitochondrial stopping mediated by PINK1 and parkin [23].

Taken together, we found that Miro1 S156A results in longer mitochondria in hDaNs; however, despite the reduction of Miro1 protein levels, mitochondrial movement is not affected under basal conditions.

### 3.2. Miro1 Degradation during CCCP-Induced Mitophagy Is Imparied by S156A Mutation

Previous studies are not conclusive on whether S156A affects Miro1 degradation upon activation of mitophagy, or if it is only relevant during low levels of PINK1/Parkin activation [15,24,26]. Hence, we next tested if Miro1 degradation is impaired. We treated hDaNs with the ionophore CCCP to induce mitophagy for different time points with the addition of the proteasome inhibitor MG132 at 6 h. In isogenic control hDaNs, we observed a decrease of Miro1 protein levels after 2 h of induction and depletion of Miro1 after 24 h of treatment (Figure 2A,B). The addition of the proteasome inhibitor did not prevent Miro1 degradation, suggesting removal via alternative pathways. However, in Miro1 S156A hDaNs, depolarization of mitochondria does not result in further Miro1 protein degradation and inhibition of the proteasome mildly increased protein levels (Figure 2A,B). After 24 h of CCCP treatment, Miro1 is depleted, comparable to the isogenic control. 

These findings indicate that the flux of Miro1 during mitophagy is impaired in S156A hDaNs.

To assess whether removal of other mitochondrial outer membrane targets of PINK1 and Parkin is impaired in Miro1 S156A hDaNs, we looked at the removal of Mitofusin. It is both ubiquitinated by Parkin [38,39] and degraded [15,38] in a similar manner to Miro1. In the isogenic control, Mitofusin degradation follows a similar degradation pattern as Miro1, but in Miro1 S156A mutant hDaNs, its degradation is slower. After 24 h, Mitofusin is depleted in both lines (Figure 2A,C). Next, we assessed levels of full-length and cleaved LC3 to measure activation of the autophagic machinery. We found that cleavage of LC3 in Miro1 S156A hDaNs occurs in a similar manner to isogenic control hDaNs (Figure 2A,D). These findings show that even though removal of Miro1 is changed in S156A hDaNs and degradation of mitochondria outer membrane targets like Mitofusin occurs slower, removal of damaged mitochondria via autophagy is not affected.

To assess whether the impairment of Miro1 degradation could be explained by a reduced sensitivity to CCCP, we looked at mitochondrial membrane potential. In live cell imaging of hDaNs stained with the mitochondrial membrane indicator TMRM and MitoTracker green as control, we found that Miro1 S156A hDaNs have a significantly reduced mitochondrial membrane potential (Figure 2E). These findings were confirmed by flow cytometry analysis of hDaNs stained with TMRM and MitoTracker deep red (Figure 2F). Because the loss of mitochondrial membrane potential could affect mitochondrial quality, we used the MitoTimer construct, which shifts from green to red fluorescence over a duration of 48 h, to assess mitochondrial turnover. The ratio between red and green fluorescence provides information about import and degradation of a mitochondrial matrix marker protein [34]. In Miro1 S156A hDaNs, the ratio between degradation (red) and import (green) is significantly decreased compared to isogenic control hDaNs (Figure 2G), indicating that the decrease of mitochondrial membrane potential is accompanied by an increase in mitochondrial turnover.

### 3.3. Miro1 S156A Reduces Mitochondrial Respiration in hDaNs

Having found that mitochondrial membrane potential is decreased concomitant with an increase in mitochondrial renewal in Miro1 S156A hDaNs, we next assessed whether this could be explained by a change in OXPHOS complexes and whether it affects mitochondrial respiration.

We found that levels of complex V (ATP5A) in Miro1 S156A hDaNs are significantly reduced by approximately 22% and complex III (UQCRC2) is reduced by approximately 26% compared to the isogenic control (Figure 3A). Complexes IV (MTCO1) and II (SDHB) are mildly reduced, and only levels of complex I (NDUFB8) remain unchanged (Appendix A). Interestingly, in hDaNs, Miro1 protein levels correlate significantly with complex V (r = 0.7910, *p* = 0.0022) and complex III (r = 0.8857, *p =* 0.0001), the correlation with complex IV protein levels is not significant (r = 0.5181, *p* = 0.0844) in hDaNs (Figure 3B).

To assess whether the reduction in membrane potential and complexes V and IV has functional consequences, we analyzed mitochondrial respiration. Miro1 S156A mutant hDaNs overall consume less oxygen (Figure 3C) and have a significant reduction in basal respiration to approximately 35% of the isogenic control (Figure 3D, left graph). However, spare respiratory capacity is only mildly decreased by 13% (Figure 3D, middle graph). This results from a concomitant decrease in maximal respiration in Miro1 S156A hDaNs (Figure 3D, right graph). 

Measurement of extracellular acidification rate (ECAR) indicates whether the cells shift their energy production to glycolysis upon reduction of mitochondrial respiration. We did not observe a shift to glycolysis in S156A hDaNs (Appendix A). These findings indicate that in S156A Miro1 hDaNs, OXPHOS is impaired without compensatory increase in glycolysis, but the response to changing energy demands (which is reflected by spare respiratory capacity) is maintained.

### 3.4. The S156A Mutation Reduces Miro1 Protein Levels Affecting OXPHOS in Postmitotic Differentiated Cells

Since S156A leads to a reduction of Miro1 protein accompanied by a reduction of OXPHOS in hDaNs, we next assessed whether this effect is due to a specific vulnerability of postmitotic neuronal cells or a mutation-specific effect. To do this, we measured Miro1 protein levels, as well as Tom 20 protein levels as an indicator for mitochondrial mass, in S156A neural precursor cells. The NPCs are (in contrast to hDaNs) mitotic, undifferentiated, and have a less complex morphology (Figure 1B and Appendix A). In untreated NPCs, Miro1 S156A does not result in a reduction in Miro1 steady-state protein levels (Figure 4A), and Tom20 protein levels are comparable between Miro1 S156A and the isogenic control NPCs (Appendix A). In NPCs, the S156A mutation alone is not sufficient to reduce OXPHOS complexes V and III (Figure 4A). OXPHOS complexes IV and I also remain unchanged (Appendix A). We then knocked down Miro1 protein in mitotic, undifferentiated SH-SY5Y neuroblastoma cells to 63% of the control (*p* = 0.0326) by transfecting siRNA pools, which had no effect on either Tom20 protein levels (Appendix A) or OXPHOS complexes (Figure 4B and Appendix A). These data support the notion that Miro1 reduction is necessary for the decrease of the OXPHOS complexes observed in hDaNs. We next performed Miro1 knockdown in post-mitotic, differentiated SH-SY5Y neuroblastoma cells to 76% of the control (*p* = 0.0039), which had no effect on Tom20 levels (Appendix A). Reduced Miro1 levels in differentiated neuroblastoma cells did significantly reduce OXPHOS complexes III and IV, but not complexes I or V (Figure 4C and Appendix A). In line with the data from hDaNs, Miro1 levels correlate with reduced mitochondrial respiratory complexes (CIII r = 0.8318, *p* = 0.0104; CIV r = 0.9630, *p* = 0.0001). These data highlight the relevance of Miro1 depletion depending on the differentiation state of neurons.

## 4. Discussion

In a previous study, the induction of mitophagy failed to trigger Miro1 degradation and/or Parkin recruitment to mitochondria in a significant number of Parkinson’s disease patients’ fibroblasts. Treatment with a Miro1 reducer compound restored Miro1 removal and translocation of Parkin to mitochondria [27]. Another study showed that Miro1 degradation is required for proper mitochondrial clearance during mitophagy [40]. Phosphorylation at Ser156 facilitates Miro1 removal under non-stimulated conditions [26]. However, the overexpression of Miro1 S156A or S156E constructs did not alter Miro1 protein levels [15]. All of these studies used different mitotic cell models such as fibroblasts, HeLa, HEK, and SH-SY5Y, and the Ser156 studies used transfection of the respective Miro1 constructs (S156A or S156E) with the endogenous protein present. Some studies additionally transfected with PINK1 and/or Parkin to assess mitophagy and mitochondrial motility. These co-transfection experiments revealed that Ser156 is important for PINK1-Parkin-mediated phosphorylation of Miro1 and the subsequent removal of Miro1 during mitophagy, but it was not clear whether Ser156 phosphorylation is required for regulating endogenous Miro1 levels.

In our model, hDaNs constitutively express S156A mutant Miro1 without wild-type Miro1 present, and we did not change the expression levels of PINK1 or Parkin by transfection of constructs. We found a significant reduction of Miro1 protein levels in Miro1 S156A hDaNs without a reduction in gene expression. Because we found no differences in mitochondrial mass, we hypothesized that the baseline reduction of Miro1 protein could be a result of either increased degradation as a compensatory mechanism or a destabilizing effect of the mutation. The fact that neural precursors with the same mutation do not have reduced Miro1 levels argues in favor of compensation during development. It is also possible that the addition of a mitochondrial stressor is needed to expose the mutation-specific effect on Miro1 steady-state levels in mitotic cells.

Previous studies have shown that chemical-induced mitochondrial depolarization in healthy cells is followed by Miro1 phosphorylation and removal [15,26,27]. In the complete absence of Miro1, mitophagy is also impaired and the ubiquitination of Miro1 was shown to be a requirement for executing mitophagy in mouse primary neurons [40]. While this indicates that Ser156 phosphorylation does not affect Miro1 ubiquitination during chemically induced mitophagy [15,24], one study indicated that this might be relevant for Parkin recruitment and moderate Parkin activation [26]. In our study, the S156A mutation inhibits the flux of CCCP-induced mitophagy marked by impaired Miro1 and delayed Mitofusin degradation, presumably because the reduced mitochondrial membrane potential and depleted Miro1 levels have already engaged mitochondrial quality control systems. Parkin targets both Miro1 and Mitofusin for degradation, and the removal of the outer mitochondrial membrane targets precedes activation of autophagy and is mainly mediated by proteasomal degradation [41]. Here, inhibition of the proteasome has little or no effect on these substrates in both S156A and the control neurons, suggesting that degradation of Miro1 and Mitofusin can occur independently of the proteasome in hDaNs. The mutation does not prevent the CCCP-induced removal of Miro1 completely since it is degraded after 24h of treatment. This can be explained by the activation of the autophagosomal system. Little is known about the effect of phosphorylation at Thr298 and Thr299 of Miro1, but a study indicates that they might oppose the effect of phosphorylation at Ser156 [26]. Since Miro1 cannot be phosphorylated at Ser156, additional phosphorylation at Thr298 and Thr299 might contribute to failure of Miro1 degradation in S156A hDaNs during the first six hours of treatment. Taken together, the findings of this study are particularly interesting since they confirm the importance of Miro1 Ser156 phosphorylation in dopaminergic neurons. This is important because degradation of Miro1 is impaired in a subset of Parkinson’s disease patients [27] and in α-synuclein mutant neurons [42].

We suggest that even a partial reduction of Miro1 steady-state levels is sufficient to impair mitophagy, but not mitochondrial movement. Together with Miro1 knockout studies in mice [40,43], our data support the notion that Miro1 has an essential role in PINK1-Parkin-mediated mitophagy. In dopaminergic neurons, the S156A mutation causes partial (40–50%) but incomplete depletion of Miro1 steady-state levels. Under physiological conditions, there are likely to be thresholds of Miro1 levels for maintaining certain functions, like mitochondrial movement, which cannot be modelled by chemically forced mitophagy or complete loss of the protein via knockout. Although Miro1 is essential for multiple processes, the whole removal of mitochondria via classical mitophagy could be the most dispensable process in mature neurons. Since previous experiments were performed in different cell types (SH-SY5Y [15], HeLa [24], and HEK/rat embryonic fibroblasts [26]), a greater significance of Ser156 phosphorylation during induced mitophagy might be attributed to the specific physiology of iPSC-derived hDaNs. It will be interesting to further detail the effect of this mutation in different cell types, including iPSCs, iPSC-derived astrocytes, or iPSC-derived cortical neurons. Furthermore, co-culturing or crossing Miro1 and PINK1 models may help in further dissecting the mechanism of Miro1 phosphorylation and subsequent events at the MOM.

Our data show a novel connection between Miro1 phosphorylation, Miro1 levels, and mitochondrial respiration. During differentiation, iPSCs switch energy production from glycolysis to OXPHOS to maintain the ATP levels required for neuronal activity [8]. The reduction of basal and maximal respiration in Miro1 S156A mutant neurons suggests that neurons might not be able to meet energetic demand during prolonged neuronal activation, which, in turn, might impair synaptic transmission. So far, only very few studies have looked at the of role Miro1 in regulating OXPHOS proteins and function. Studies in Miro1 knockout mouse fibroblasts (MEFs) found no effect on mitochondrial respiration [7,44], indicating that depletion of Miro1 does not alter their mitochondrial bioenergetics. In line with these findings, we showed that in undifferentiated SH-SH5Y cells, knockdown of Miro1 does not reduce complex IV and V protein levels. S156A might also impair Miro1 interaction with Parkin [23,26], and it is possible that PINK1/parkin signaling might confer the effect on respiration and levels of respiratory complexes since PINK1 and Parkin regulate synthesis of OXPHOS complexes [45]. We hypothesize that as part of mitochondrial quality control, phosphorylation of Miro1 at Ser156 promotes the synthesis of OXPHOS subunits or their regulation via Miro1 steady-state levels.

## Figures and Tables

**Figure 1 cells-11-01269-f001:**
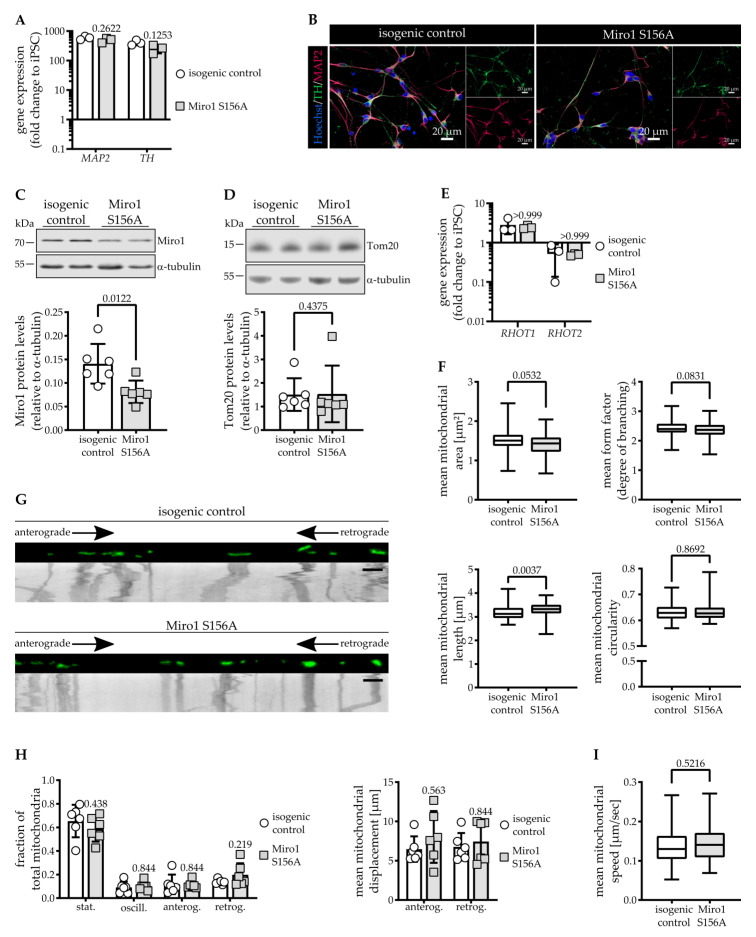
Reduction of Miro1 protein levels in Miro1 S156A hDaNs. (**A**) RT-qPCR showing expression of dopaminergic neuronal markers in hDaNs as fold change to iPSCs. Error bars depict SD (n_Diff_ = 3); 2way ANOVA with Tukey’s multiple comparisons. (**B**) Representative immunofluorescence staining showing the mature neuronal marker MAP2 and dopaminergic marker TH in hDaNs (n_Diff_ = 3). (**C**) Representative Western blot showing Miro1 in hDaN lysate. Quantification of intensity of Miro1 Western blot bands normalized to α-tubulin. Error bars depict SD (n_Diff_ = 6); paired *t* test, two-tailed. (**D**) Representative Western blot showing Tom20 in hDaN lysate. Quantification of intensity of Tom20 Western blot bands normalized to α-tubulin. Error bars depict SD (n_Diff_ = 6); Wilcoxon test matched pairs signed rank test. (**E**) RT-qPCR showing gene expression of *RHOT1/2* in hDaNs as fold change to iPSCs. Error bars depict SD (n_Diff_ = 3); Wilcoxon matched pairs signed rank test with two-stage linear step-up procedure of Benjamini, Krieger, and Yekutieli. (**F**) Imaging analysis of mitochondrial morphology in hDaNs stained with 100 nM MitoTracker green. Error bars depict SD (n_Diff_ = 3). Mitochondrial area and from factor: unpaired *t* test, two-tailed; Mitochondrial length and circularity; Mann-Whitney test. (**G**) First frame of representative movie with corresponding kymograph below for analysis of mitochondrial movement in processes of hDaNs stained with 100 nM MitoTracker green. Scale bar represents 10 µm. (**H**) Kymograph analysis of stationary, oscillating, anterograde, and retrograde mitochondrial fractions and mean displacement of anterograde and retrograde moving mitochondria. Error bars depict SD (n_Diff_ = 3); Wilcoxon matched pairs signed rank test with two-stage linear step-up procedure of Benjamini, Krieger, and Yekutieli. (**I**) Analysis of mitochondrial mean speed using Fiji plugin TrackMate. Error bars depict SD (n_Diff_ = 3); Mann-Whitney test.

**Figure 2 cells-11-01269-f002:**
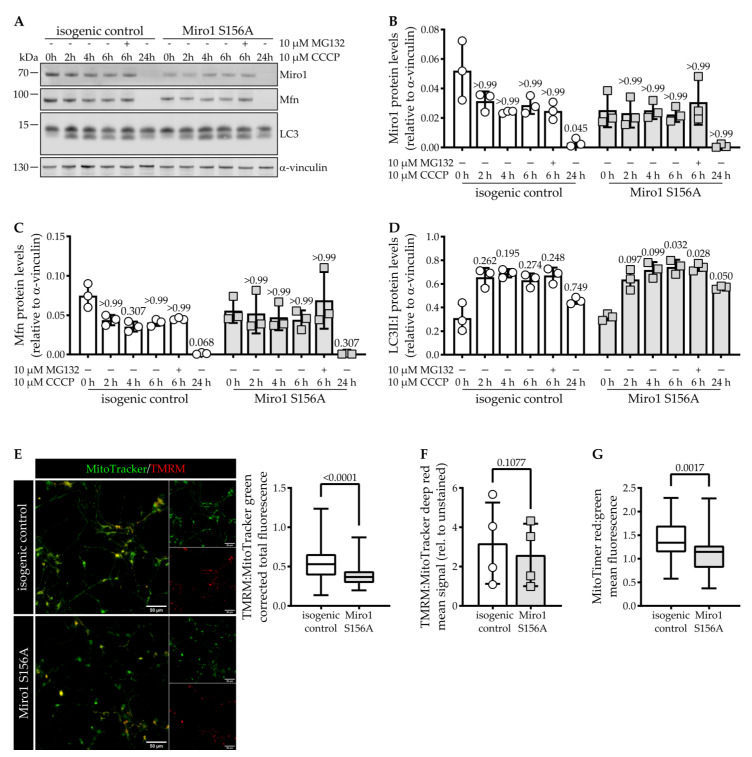
Miro1 degradation is impaired in Miro1 S156A hDaNs. (**A**) Representative Western blot showing Miro1, Mitofusin, and LC3 in hDaN lysate. Neurons were treated with a final concentration of 10 µM CCCP for 0, 2, 4, 6 (±10 µM MG132), or 24 h to induce mitophagy. (**B**) Quantification of intensity of Miro1 Western blot bands normalized to α-vinculin. Error bars depict SD (n_Diff_ = 3); Friedman’s test with Dunn’s multiple comparisons relative to 0 h of the respective genotype. (**C**) Quantification of the intensity of Mitofusin Western blot bands normalized to α-vinculin. Error bars depict SD (n_Diff_ = 3); Friedman’s test with Dunn’s multiple comparisons relative to 0 h of the respective genotype. (**D**) Ratio of LC3II:LC3I quantification of the intensity of LC3 Western blot bands normalized to α-vinculin. Error bars depict SD (n_Diff_ = 3); 2way ANOVA with Tukey’s multiple comparisons. (**E**) Imaging analysis of mitochondrial membrane potential in hDaNs stained with 25 nM TMRM and 100 nM MitoTracker green. Representative live cell image of TMRM and MitoTracker green staining. Quantification of corrected total fluorescence of TMRM relative to MitoTracker green. Error bars depict SD (n_Diff_ = 3); Mann-Whitney test. (**F**) Flow cytometric measurement of mitochondrial membrane potential in hDaNs stained with 100 nM TMRM and 100 nM MitoTracker deep red. Mean TMRM signal relative to MitoTracker deep red normalized to unstained signal. Error bars depict standard deviation (n_Diff_ = 4); paired *t* test, two-tailed. (**G**) Imaging analysis of mitochondrial renewal in hDaNs using MitoTimer plasmid. Quantification of mean red fluorescence relative to mean green fluorescence. Error bars depict SD (n_Diff_ = 3); Mann-Whitney test.

**Figure 3 cells-11-01269-f003:**
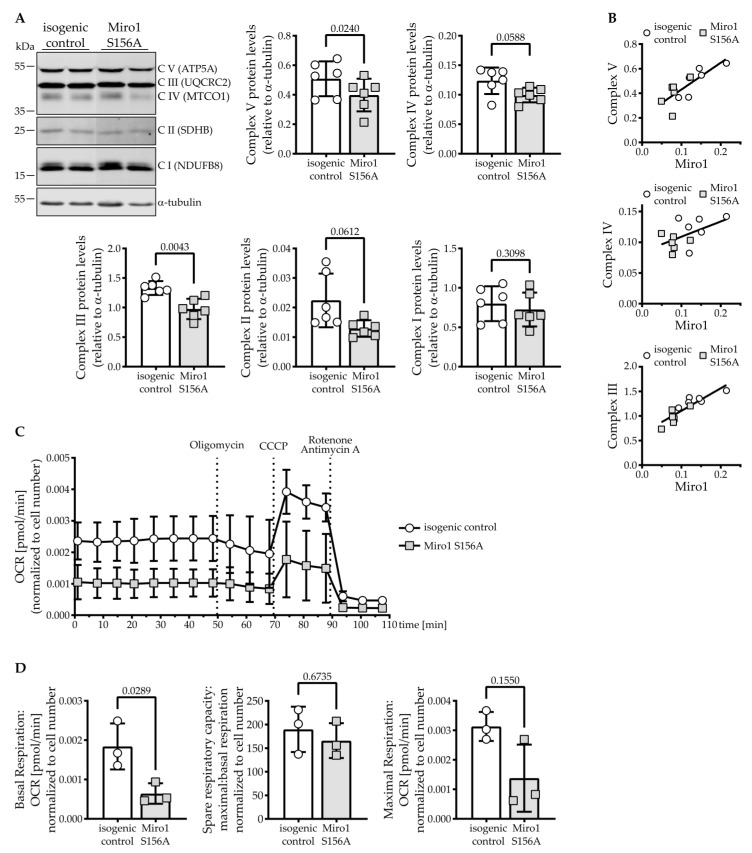
Miro1 S156A reduces mitochondrial respiration in hDaNs. (**A**) Representative Western blot showing complex V/IV/III/II/I in hDaN lysate. Quantification of the intensity of bands normalized to α-tubulin. Error bars depict standard deviation (n_Diff_ = 6); paired *t* test, two-tailed. (**B**) Correlation of Miro1 protein levels to complex V/IV/III protein levels in hDaN lysate (n_Diff_ = 6). (**C**) Respiratory analysis of mitochondrial oxygen consumption in hDaNs. Neurons were challenged with Oligomycin, CCCP, and rotenone with antimycin A at the indicated times. Oxygen consumption rate was normalized to the number of cells seeded. Error bars indicate standard deviation (n_Diff_ = 3). (**D**) Basal respiration, spare respiratory capacity, and maximal respiration calculated from respiratory analysis. Error bars indicate standard deviation (n_Diff_ = 3); paired *t* test, two-tailed.

**Figure 4 cells-11-01269-f004:**
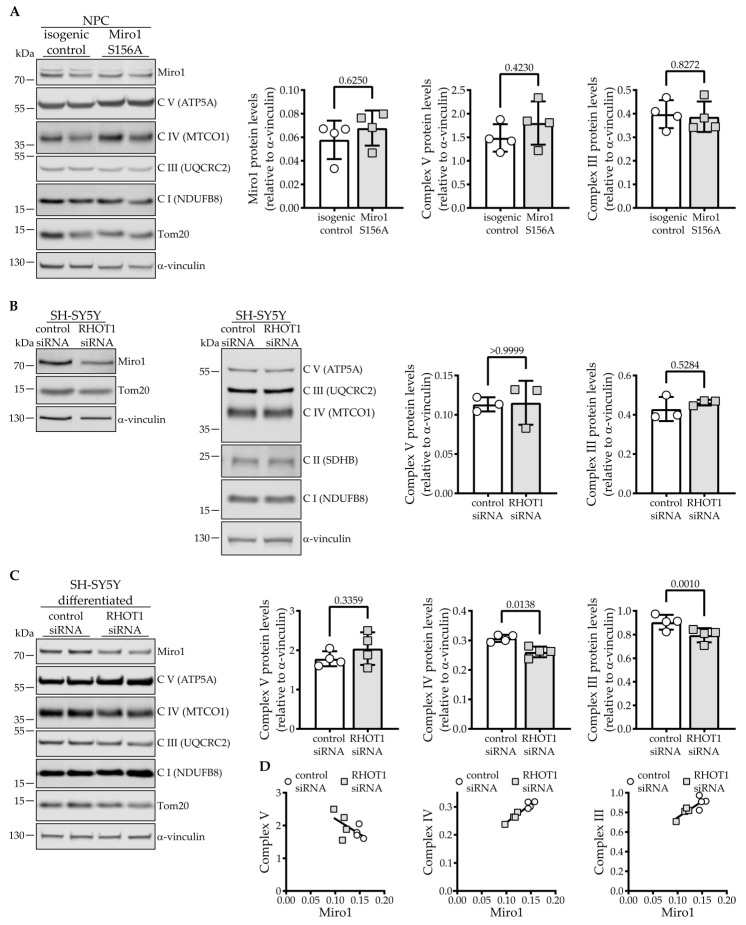
The S156A mutation reduces Miro1 protein levels affecting OXPHOS in postmitotic differentiated cells. (**A**) Representative Western blot showing Miro1, Tom20, complex V/IV/III/I in NPCs. Quantification of intensity of Miro1, complex V and III bands normalized to α-vinculin. Error bars depict standard deviation (n = 4); Miro1: Wilcoxon test matched pairs signed rank test; complexes V and III: paired *t* test, two-tailed. (**B**) RHOT1 knockdown in undifferentiated SH-SY5Y cells. Representative Western blot showing Miro1, Tom20, complex V/IV/III/II/I in undifferentiated SH-SY5Y cells transfected with RHOT1 or non-targeting siRNA. Quantification of intensity of complex V and III bands normalized to α-vinculin. Error bars depict standard deviation (n = 3); complex V: Wilcoxon test matched pairs signed rank test.; complex III: paired *t* test, two-tailed. (**C**) RHOT1 knockdown in differentiated SH-SY5Y cells. Representative Western blot showing Miro1, Tom20, complex V/IV/III/I in differentiated SH-SY5Y cells transfected with RHOT1 or non-targeting siRNA. Quantification of intensity of complex V/IV/III bands normalized to α-vinculin. Error bars depict standard deviation (n = 4); paired *t* test, two-tailed. (**D**) Correlation of Miro1 protein levels to complex V/IV/III protein levels in differentiated SH-SY5Y cells (n = 4).

**Table 1 cells-11-01269-t001:** Primers used for RT-qPCR.

Gene	Forward	Reverse
GAPDH	CGAGATCCCTCCAAAATCAAG	GCAGAGATGATGACCCTTTTG
NESTIN	GGCAGCGTTGGAACAGAGGTTGGA	CTCTAAACTGGAGTGGTCAGGGCT
SOX2	TTCACATGTCCCAGCACTACCAGA	TCACATGTGTGAGAGGGGCAGTGTGC
PAX6	GTGTCCAACGGATGTGTGAG	CTAGCCAGGTTGCGAAGAAC
TH	TGTCTGAGGAGCCTGAGATTCG	GCTTGTCCTTGGCGTCACTG
MAP2	CCGTGTGGACCATGGGGCTG	GTCGTCGGGGTGATGCCACG
RHOT1	TGTCACCCCAGAGAGAGTTC	GCCTGCTGTCTTTGTCTGTT
RHOT2	ATTGAGACCTGCGTGGAGTG	AAGCGTTGAGCTCTTCGTCA

## Data Availability

Not applicable.

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
