# Peer review of "Steady-State Levels of Miro1 Linked to Phosphorylation at Serine 156 and Mitochondrial Respiration in Dopaminergic Neurons"

_cells, 2022, doi:10.3390/cells11081269_

Round 1

Reviewer 1 Report

Schwarz & Fitzgerald present here an excellent study on the consequences of mutation of S156 of Miro1 (as a model of non-phosphorylation) and its consequences on dopaminergic neurons, most of which are carried out in neurons derived from IPSCs. The technical standard of the paper is outstanding, and a I commend the authors for their approach and thoroughness. I only have a few concerns, all related to presentation:

  • The study relies on two lines of IPSC-derived dopaminergic neurons, one with the S156A mutation, one control, which the authors have characterised in previous papers. Given how critical these are to the manuscript, however, I feel that it is important that the authors also provide some information on their characterisation here as well, to confirm to readers that the two are directly comparable.
  • The title begins that “Phosphorylation of Miro at Serine 156…”, however the experiments in the paper focus only on mutation of S156 as a model. I agree that it is most likely that effects of S156A derive from the lack of phosphorylation, but this is not specifically demonstrated in this paper. I would therefore prefer a toned down title that states that the S156A has effects, which implicates phosphorylation.

Minor points:

  • Western blots should be “wider” with the equivalent of 4-5 band widths above and below each signal, to confirm lack of other bands to the reader. This is purely a presentation point/reader service, however, as the authors also supply full blots as supplementary material
  • Personally I find the combining of results and discussion into a single section to be less than ideal, as the subchapters mean that there are effectively several miniature discussions. However, this is a matter of journal editorial policy, and therefore their decision.

Author Response

Thank you for taking the time to review our manuscript. We found the reviewer comments very useful and have now made significant changes to the manuscript which we have resubmitted and highlighted in grey. We have also responded point by point to each of the comments below.

  • The study relies on two lines of IPSC-derived dopaminergic neurons, one with the S156A mutation, one control, which the authors have characterised in previous papers. Given how critical these are to the manuscript, however, I feel that it is important that the authors also provide some information on their characterisation here as well, to confirm to readers that the two are directly comparable.

Response: Thank you this is a valid point. We therefore include more information at the beginning of the first results section and refer directly to the comparability of the cell lines.

  • The title begins that “Phosphorylation of Miro at Serine 156…”, however the experiments in the paper focus only on mutation of S156 as a model. I agree that it is most likely that effects of S156A derive from the lack of phosphorylation, but this is not specifically demonstrated in this paper. I would therefore prefer a toned down title that states that the S156A has effects, which implicates phosphorylation.

Response: Thank you for this suggestion and we agree. We have therefore changed the title to “Steady-state levels of Miro1 linked to phosphorylation at Serine 156 and mitochondrial respiration in dopaminergic neurons”.

Minor points:

  • Western blots should be “wider” with the equivalent of 4-5 band widths above and below each signal, to confirm lack of other bands to the reader. This is purely a presentation point/reader service, however, as the authors also supply full blots as supplementary material.

Response: Thank you very much, we agree and this was no problem for us to correct. We have now re-cropped the Western blots in the figures and the cropping area and the original blots have been submitted as a separate Word document.

  • Personally I find the combining of results and discussion into a single section to be less than ideal, as the subchapters mean that there are effectively several miniature discussions. However, this is a matter of journal editorial policy, and therefore their decision.

Response: Thank you, we agree. We have now re written and re formatted the manuscript so that there is a separate results and discussion section.

Reviewer 2 Report

In light of the fact that it is a target of both PINK 1 and Parkin, Miro1 has emerged as an interesting target for studying Parkinson's disease-related pathways. The purpose of this manuscript was to investigate the effect of the RHOT1/Miro1 S156A mutation on the degradation of Miro1 and cellular respiration. It has been suggested that Ser156 is a PINK1 phosphorylation site and involved in the degradation of Miro1 and mitophagy arrest.

Using gene-edited Miro1 S156A dopaminergic neurons, authors have compared their function to an isogenic control. Miro1 steady-state protein levels are reduced approximately 50% by homozygous Ser156 deletion, resulting in impaired mitophagy upon CCCP activation. The mutation causes an abnormal reduction in mitochondrial oxygen consumption under basal conditions. This is accompanied by a decrease in OXPHOS complexes III and V. However, Miro1 siRNA knockdown did not produce the same results in undifferentiated, mitotic cells. This suggests phosphorylation in mature neurons is especially relevant.

Taken together, in mature dopaminergic neurons, inhibition of PINK1 phosphorylation at Ser156 does affect mitochondrial energetics, but it does not alter mitochondrial area, shape, or lifespan. Using these findings, treatment strategies that target or reduce Miro1 could be improved for Parkinson's disease.

In general, I found the manuscript very appealing to read. The authors have presented the manuscript with a clear storyline, though the manuscript needs extensive language editing. However, I have the following concerns.

  1. Authors have shown SH-SY5Y knockdown of Miro1 does not reduce complex IV and V protein levels, which is similar to mouse Miro1 fibroblast knockout. Though the role appears very neuronal-specific, some experimental evidence is still missing. Authors could demonstrate the neuronal-specific contribution of Miro1 on mitochondrial respiration on neurons differentiated from the SH-SY5Y cell line.

This would be considered as an ideal control.

  1. Authors should supplement siRNA sequences which use in this study.
  2. The discussion has to be improved moderately in context to the available Miro1 knockout studies.
  3. Authors should take care of typo/spelling errors.

Author Response

Thank you for taking the time to review our manuscript. We found the reviewer comments very useful and have now made significant changes to the manuscript which we have resubmitted and highlighted in grey. We have also responded point by point to each of the comments below.

  • In general, I found the manuscript very appealing to read. The authors have presented the manuscript with a clear storyline, though the manuscript needs extensive language editing. However, I have the following concerns.

Response: Thank you for your suggestion. We agree and have now extensively re-written and re-edited the new manuscript.

  • Authors have shown SH-SY5Y knockdown of Miro1 does not reduce complex IV and V protein levels, which is similar to mouse Miro1 fibroblast knockout. Though the role appears very neuronal-specific, some experimental evidence is still missing. Authors could demonstrate the neuronal-specific contribution of Miro1 on mitochondrial respiration on neurons differentiated from the SH-SY5Y cell line. This would be considered as an ideal control.

Response: Thank you very much for this suggestion. We have performed these experiments in several independent differentiations of SH-SY5Y cells. We were successful in achieving significant Miro1 knockdown in the differentiated cells and were able to show that indeed the differentiation state of the cells is important. In parallel to these experiments, we also performed Western blotting on lysates from untreated, undifferentiated neural precursor cells harbouring the S156A mutation. Overall these data show that baseline reduced OXPHOS complex levels correlate to Miro1 levels and that the effect can only be observed in post mitotic cells highlighting the vulnerability of differentiated neurons to Miro1 S156A. Our new data remains in line with our previous hypotheses and with the work of other researchers in the Miro1 field. We think that this data now further strengthens the relevance of this mitochondrial quality control pathway in Parkinson’s disease.

  • Authors should supplement siRNA sequences which use in this study.

Response: Thank you for the suggestion. We checked whether the sequences are available from Dharmacon. The sequences are not published since the company sells the product as a ‘SMART Pool’and not individual siRNAs. We have good experience with Dharmacon SMART Pools for efficient knockdown so we instead refer to the siRNAs being pooled in the text and highlight the company and catalogue number. We hope this suffices.

  • The discussion has to be improved moderately in context to the available Miro1 knockout studies.

Response: Thank you, we agree. The discussion was modified extensively and we included more discussion on Miro1 levels including its knockout. We also refer to recent work in the KO mouse models from the Kittler lab and several other publications. We hope this now suffices.  

  • Authors should take care of typo/spelling errors.

Response: Thank you, we have checked over the new manuscript several times and asked our colleagues to check for typos and spelling mistakes. We hope that we have corrected them all.

Round 2

Reviewer 2 Report

  Best wishes for the authors' future research work.